# Tracing the Origin of Planktonic Protists in an Ancient Lake

**DOI:** 10.3390/microorganisms8040543

**Published:** 2020-04-09

**Authors:** Nataliia V. Annenkova, Caterina R. Giner, Ramiro Logares

**Affiliations:** 1Limnological Institute Siberian Branch of the Russian Academy of Sciences 3, Ulan-Batorskaya St., 664033 Irkutsk, Russia; 2Institute of Marine Sciences (ICM), CSIC, Passeig Marítim de la Barceloneta, 37-49, ES08003 Barcelona, Spain; caterina@icm.csic.es; 3Institute for the Oceans and Fisheries, University of British Columbia, 2202 Main Mall, Vancouver, BC V6T 1Z4, Canada

**Keywords:** microbial eukaryotes, phylogeography, marine-freshwater transitions, evolutionary radiation, species flocks, Lake Baikal, sympatry

## Abstract

Ancient lakes are among the most interesting models for evolution studies because their biodiversity is the result of a complex combination of migration and speciation. Here, we investigate the origin of single celled planktonic eukaryotes from the oldest lake in the world—Lake Baikal (Russia). By using 18S rDNA metabarcoding, we recovered 1414 Operational Taxonomic Units (OTUs) belonging to protists populating surface waters (1–50 m) and representing pico/nano-sized cells. The recovered communities resembled other lacustrine freshwater assemblages found elsewhere, especially the taxonomically unclassified protists. However, our results suggest that a fraction of Baikal protists could belong to glacial relicts and have close relationships with marine/brackish species. Moreover, our results suggest that rapid radiation may have occurred among some protist taxa, partially mirroring what was already shown for multicellular organisms in Lake Baikal. We found 16% of the OTUs belonging to potential species flocks in Stramenopiles, Alveolata, Opisthokonta, Archaeplastida, Rhizaria, and Hacrobia. Putative flocks predominated in Chrysophytes, which are highly diverse in Lake Baikal. Also, the 18S rDNA of a number of species (7% of the total) differed >10% from other known sequences. These taxa as well as those belonging to the flocks may be endemic to Lake Baikal. Overall, our study points to novel diversity of planktonic protists in Lake Baikal, some of which may have emerged in situ after evolutionary diversification.

## 1. Introduction

Even though single cell eukaryotes are fundamental for ecosystem function, a substantial fraction of their diversity remains poorly known. During the last decade, multiple studies have discovered new protist diversity, which was in part possible due to culture-independent molecular approaches using high-throughput sequencing (HTS) and metabarcoding [1,2]. Metabarcoding is highly relevant for pico/nano-sized (~0.8–5 µm, as defined in [3]) unicellular eukaryotes as they are too small to show unambiguous morphological differences and are difficult to culture [3,4]. The key ecological role of these minute eukaryotes has been shown in both oceans [3,5] and lakes [6,7]. Yet, we are still beginning to understand their diversity, evolution, and dispersal.

In contrast to marine protists, we know less about the diversity of freshwater counterparts. Pico/nano-sized freshwater protists have been studied using Sanger sequencing in European ponds, lakes [8,9], and Lake George in the USA [10]. More recently, small protists were investigated using metabarcoding in various lakes of Western Europe [11,12], as well as a few lakes in North [13] and South America [14]. Most of these studies pointed to diverse patterns of community structure and novel diversity. Yet, the previous reflects another bias: most studies of freshwater protists correspond mainly to Northern Europe and North America [15]. In contrast, other continents, like Asia, remain underexplored in terms of freshwater protist diversity, apart from a few lakes in China [16] and in the high-mountain Himalayan [17].

Lake Baikal (Siberia, Northern Asia) is the oldest (more than 26 million years), largest (23,615 km^3^), and deepest (down to 1642 m depth, with a mean depth of 744 m) freshwater lake in the world (Figure 1). As a comparison, the oldest basin of Lake Tanganyika (Africa) is 9–12 million years old and the Great Lakes in North America began to form around 14,000 years ago. Baikal present-day communities inhabit the whole water-column and include more than 3500 known species of micro- and macro-eukaryotic organisms, of which more than half seem to be endemic [18]. Yet, this diversity likely represents only a fraction (perhaps 60%) of the total diversity of Lake Baikal [18] as every year new species are being described. Such biodiversity is the result of a complex process including in situ evolutionary diversification as well as immigrations or extinctions in part related to geologic activity and climate change. Few relict groups are 30–70 M years old, while other organisms appeared 3.5–30 M years ago during the isolation of the Baikal depression. Furthermore, various taxa explosively developed 0.15–3.5 M years ago in response to the most dramatic tectonic changes in the region coupled with climate cooling [19]. One of the common mechanisms that led to speciation within multicellular organisms in Lake Baikal was adaptive radiation, which resulted in endemic species flocks [18,20].

Compared to its eukaryotic macrobiota, much less is known about the diversity, evolution, and ecology of Lake Baikal eukaryotic microbiota. Routine monitoring of Lake Baikal plankton has been carried out using light microscopy for more than 60 years [21,22,23]. These studies showed seasonally recurrent changes in phytoplankton as well as long-term changes that may be explained by global warming. Furthermore, the previous studies pointed to phytoplankton spatial heterogeneity in spring, also indicating a high productivity of picophytoplankton (up to 60% of phytoplankton biomass in some years). Microscopy studies of Baikal protists indicated both cosmopolitan freshwater species as well as examples of endemicity [24,25]. More detailed studies of diatoms [26] and dinoflagellates [27] revealed the existence of species flocks. Recently, two studies have investigated the diversity of protist communities in Lake Baikal using metabarcoding. The first study [28] provided insights on the genetic diversity of protists from the southern basin of Lake Baikal (see Figure 1), pointing to novel diversity. Another study investigated co-occurrence networks of bacteria and microbial eukaryotes during the spring season [29], indicating positive correlations between specific Operational Taxonomic Units (OTUs) as well as the influence of geography on the microbial community structure. Despite the insights provided by these studies, more efforts are needed in order to understand protist diversity and evolution in Lake Baikal. The main reasons are related to the huge lake size, which provides many different niches in upper or deeper waters, open waters or bays, as well as the lake’s age, which, different to most widely-studied younger lakes, has allowed evolutionary diversification to take place in situ. Such evolutionary analyses, so far barely considered in previous studies, should provide important information about endemism in Lake Baikal protists as well as protist phylogeography in fresh waters.

The aim of this study was to investigate the origin of the planktonic protists populating the pico/nano size fraction (0.45–8 µm in size) in Lake Baikal. To do so, we analyzed the phylogeographic patterns, in particular, the evolutionary closeness of Baikal protists to known freshwater or marine taxa from all over the world as well as the potential existence of evolutionary radiations that may have led to species flocks and endemicity.

## 2. Methods

### 2.1. Sample Collection, DNA Extraction, and Sequencing of 18S rRNA Genes

Plankton samples were collected in July 2013 using a Niskin bottle from 14 sites in Lake Baikal, Russia (Figure 1). In three sites located near the coast (depth was less than 40 m), samples were taken at 5 m depth and in one coastal site (1CC) samples were taken at 1 and 20 m depth and mixed. The rest of the sites were located in open Baikal waters and samples were taken at 1, 15, and 50 m depth. These sampling depths are typically used for surface samples in Lake Baikal. Two samples were taken at 25 m depth instead of 15 m due to sampling logistics. Water samples were kept in dark and cold conditions before filtration. To collect microbial DNA, up to 5 L of water was pre-filtered through a 70 μm mesh-size net to remove larger organisms and then sequentially filtered through 8 μm and 0.45 μm polycarbonate membrane filters (Sartorius, Göttingen, Germany) using a vacuum pump. Filters were immediately frozen at −20 °C and stored at −70 °C until DNA extraction. Only 0.45 μm pore-size filters were used for DNA extraction in order to focus in small protists (cell size 0.45–8 μm).

DNA was extracted from one filter per site using the DNeasy Plant Kit (QIAGEN GmbH, Hilden, Germany) following the manufacturer’s instructions. The eukaryotic universal primers TAReukFWD1 (5′-CCAGCASCYGCGGTAAT) and TAReukREV3 (5′-ACTTTCGTTCTTGATYRA-3′) were used for PCR amplification of the 18S RNA gene (V4 region; ~380 bp) [30]. Amplicons from 3 independent PCR reactions for each sample were pooled together. PCR products were sequenced at Fasteris, Switzerland (https://www.fasteris.com) on an *Illumina* MiSeq (2 × 250 bp). Overall, 23 samples were successfully sequenced.

### 2.2. V4-Amplicon Data Processing and Statistical Analyses

Reads were processed following an in-house pipeline [31]. Briefly, raw reads were corrected using BayesHammer [32] following [33]. Corrected paired-end reads were subsequently merged with PEAR [34] and sequences > 200 bp were quality-checked (maximum expected errors 0.5) and de-replicated using USEARCH [35]. OTUs were delineated at 97% and 99% similarity using UPARSE V8.1.1756 [36]. We used the 97% cut-off to compare the number of OTUs from our study against other studies of protists using this cut-off. However, all downstream analyses used a 99% cut-off given that it provides a higher taxa definition than the commonly-used 97% cut-off, also being less prone to reflect population variation as could be the case when using sequence variants [37]. To obtain OTU abundances, reads were mapped back to OTUs at 99% similarity using an exhaustive search (*-maxaccepts 20 -maxrejects 100,000*). Chimera check and removal was performed both de novo and using the SILVA reference database [38]. Before the removal of metazoan and nucleomorph sequences, the number of sequences per sample ranged between 47,195 and 142,260, with a mean of 99,307. Taxonomic assignment of OTUs was done by BLASTing [39] OTU-representative sequences against four reference databases, SILVA123 [37], PR^2^ [40], and two in-house marine protist databases [41]), with one based on a collection of Sanger sequences from molecular surveys [MAS database] [42] and the other based on 454 reads from the BioMarKs project [43]. BLAST hits were filtered prior to taxonomy assignment using an in-house Python script [44], considering a coverage > 70%, a minimum alignment length of 200 bp, and an e-value < 0.00001. OTUs with representative sequences displaying a percentage of identity < 90% against their best BLAST hits were taxonomically assigned based on phylogenetic analyses. Metazoan and nucleomorph OTUs were excluded from the analyses. Computing analyses were performed at the MARBITS bioinformatics platform of the Institut de Ciències del Mar (ICM; http://marbits.icm.csic.es). Sequences are publicly available at the European Nucleotide Archive (accession number PRJEB24415) [45].

Statistical analyses of the communities were run in the R environment [46] using the package Vegan. To allow for comparisons between samples, the OTU table was randomly subsampled to the minimum number of reads per sample (33,735 reads) using the rrarefy function in the *Vegan* package. Bray-Curtis dissimilarities were used as an estimator of beta-diversity. Differences in community composition between samples were analysed using NMDS in *Vegan*. Venn diagrams were constructed with the VennDiagram package. The percentage of nucleotide differences was used to define high or low similarities between OTUs and other reported sequences. Those OTUs that differed ≤ 1% from other reported sequences were regarded as similar. OTUs that differed from known DNA sequences ≤ 10% were considered as taxa with unclear distribution: they could belong to both widespread or endemic organisms. Then, OTUs that displayed > 10% difference from any known DNA sequence were regarded as potential endemics of the Baikal region. To search for putative cases of species radiations, we used the following criteria. Three or more 99% similarity OTUs should form one clade with bootstrap support > 60%. OTUs from such clades should have >97% identity on average. These criteria aimed to reflect features of adaptive radiations such as common ancestry and rapid speciation [47]. Also, the abundance between at least three OTUs within such clades should differ < 100 times, otherwise the variation was regarded as intragenomic.

### 2.3. Phylogenetic Analysis

All Baikal OTUs were classified into six superphylums (Stramenopiles, Alveolates, Rhizaria, Hacrobia, Opisthokonta, Archaeplastida) according to BLAST analyses using SILVA and MAS databases. When downstream phylogenetic analyses showed uncertain placement of an OTU, we manually rechecked its superphylum affiliation. Sequences were aligned using MAFFT with E-INS-i strategy [48] and were manually checked. TrimAl [49] was run to remove ambiguous regions from the alignment. Maximum Likelihood (ML) phylogenetic trees based on the generated alignments were inferred using RAxML with the GTR-CAT model and default parameters [50]. Bootstrap values were estimated with 1000 pseudoreplicates using the rapid bootstrapping algorithm.

To construct phylogenetic trees, we used Lake Baikal OTUs, closely related sequences from NCBI-nr, other well-known members of each phylum, and environmental OTUs from European shallow freshwater environments from [12]. In the first step of the phylogenetic analysis, Baikal OTUs and OTUs from [12] (marked as OTU_EU_Pond) were used. Those OTUs from [12], which did not cluster with Baikal sequences, were excluded. In the second step, the closest BLAST hits as well as representatives of all classes from each phylum were added to the analysis. Those representatives, which did not have any relationships with Baikal OTUs (e.g., raphidophytes), were excluded. In the last step, additional members of classes, which showed similarity with Baikal OTUs, were added.

## 3. Results

### 3.1. Protist Composition

A total of 2,284,061 quality-filtered reads were obtained from 23 water samples (Figure 1). They grouped into 1414 protistan and 37 metazoan OTUs delineated using a cut-off value of 99% of sequence identity. The metazoans belonged to known Baikal multicellular organisms (mainly copepoda). The protistan OTUs were analysed according to their relative abundance [51]. In total, 24 OTUs were abundant (>1% of all reads), 1129 OTUs were rare (<0.01%), and 317 OTUs had intermediate abundances (0.01% < OUT < 1%). Among the rare OTUs, only 344 OTUs were represented by less than 10 reads, while others were represented by up to 230 reads. The rarefaction curves (Appendix A) indicated that we approached diversity saturation.

The phylogenetic diversity of protist OTUs covered Opisthokonta, Archaeplastida, SAR (Stramenopiles–Alveolata–Rhizaria), and taxa of uncertain positions that were combined in Hacrobia (Cryptophyta, Centroheliozoa, Telonemida, Haptophyta) (Figure 2). Stramenopiles were the most diverse group (specially Chrysophyceae), encompassing 562 OTUs (39% of all protist OTUs, Figure 2), being the second in abundance. Alveolates were the second most diverse, but the most abundant supergroup (up to 72% of reads per sample). However, their diversity may be overestimated due to the contribution of *Gyrodinium*-like sequences, which represented 40% of all alveolate sequences (Figure 2). Among them, one OTU was represented by 214,236 reads in total and had 100% similarity with the known Baikal species *Gyrodinium helveticum*. The remaining 191 OTUs had less than 100 reads and had 95–99% similarity with *G. helveticum*. They likely represent variants of the rRNA gene as dinoflagellates normally have many 18S rDNA copies per genome [52].

The prevalence of the different taxa changed when considering Operational Taxonomic Units (OTU) richness or abundance (Figure 3). In particular, Eustigmatophyceae was represented only by two OTUs, but they corresponded to 3.3% of all reads. Prasinophytes had a similar abundance (3.1% of all reads), but they were more diverse (40 OTUs). Perkinsiids, Bicosoecida, Bolidophyceae, MAST (Marine stramenopile), and some fungi (Chytridiomycota and Cryptomycota) were rare in the samples, but included many different OTUs (Figure 3).

### 3.2. Phylogenetic Relationships

About 60% of Baikal OTUs clustered with environmental DNA sequences from diverse freshwater bodies (mainly lakes) with good statistical support (>60% Bootstrap Support). These analyses indicated that Lake Baikal is populated by many protists that are evolutionary distantly related to cultured species, being instead related to unclassified microorganisms from diverse locations.

Interestingly, members of the so far exclusively-marine family Pelagophyceae were found in Lake Baikal: two closely related OTUs (corresponding to 547 and 25 reads, respectively) clustered with marine *Ankylochrysis lutea* and two Arctic *Pelagophyceae* sp. (ML bootstrap support = 100) (Figure 4, Appendix A). Specific taxa that are normally absent or infrequent in freshwater lakes were also found in Lake Baikal (Figure 4). In particular, a well-defined clade was composed by members of syndiniales and one Baikal OTU (Figure 4, Appendix A). The Baikal representative was very similar (99%) to an uncultured Amoebophryan from the brackish Arctic Ward Hunt Lake. This OTU was sequenced in four different stations in the Central Basin of Lake Baikal (totaling 308 reads).

Among Stramenopiles, members of four MAST groups were found (Figure 4, Appendix A). Baikal members of MAST-3J and MAST-6 were similar to sequences from a pond in the Natural Regional Park (France). Baikal MAST-2A was identical to a previously sequenced clone from Baikal (JN547283) and identical also to a sequence from Lake Kusaki (Japan). MAST-12 was more diverse and included 11 OTUs from MAST-12C and three OTUs from MAST-12A and -12B (Figure 4, Appendix A). There were several OTUs that could also belong to MAST: one OTU clustered with MAST-6 and three OTUs clustered with MAST-24 with low bootstrap support (Appendix A). Labyrinthulomycetes, which are mostly marine protists, were represented in Baikal by two OTUs, one of which was similar to an unclassified freshwater sequence from a French pond and another one that clustered with a sequence from a lake in Villerest (France) (Figure 4, Appendix A). A few OTUs had similarity with the Pirsonia clade, but bootstrap support was only 58 (Figure 4, Appendix A). Bolidophyceae, which are common in the ocean but mostly absent in freshwater environments [53], were represented by 10 OTUs in Lake Baikal (Figure 4, Appendix A). Lastly, two OTUs clustered with strong statistical support with marine Cercozoa from the Mataza group (Figure 4, Appendix A).

Besides high rank taxa, “marine elements” at the genus/species level were identified in Lake Baikal. In particular, two Baikal ciliates (OTU_27, 16,574 reads and OTU_579, 25 reads) and one green algae (OTU_532, 35 reads) were very similar to counterparts from the Baltic Sea (Appendix A). Also, OTU_80 (3848 reads) from Lake Baikal was identical to the sequence of an unclassified ciliate from the Bering Sea, while another Baikal ciliate (OTU_1062, 6 reads) was similar to an environmental marine sequence collected near Norway (Appendix A). One of the Telonema-like OTUs (OTU_73) clustered exclusively with *Telonema antarcticum* and several uncultured marine eukaryotes (Appendix A). Among them, OTUs from the brackish Arctic Ward Hunt Lake were identical to the OTU_73 from Lake Baikal, which was found in 18 samples and corresponded to 0.2% of all reads in Lake Baikal (4495 reads in total). The diatom *Thalassiosira pseudonana*, known from brackish waters [54], was also found in our data (Appendix A). Furthermore, OTU_299 (155 reads; Appendix A) was identical to *Naganishia vishniacii* (*Cryptococcus vishniacii*), which is an enigmatic protist from Antarctic soil [55] but was also found in the meltwater from Tianshan glacier [56].

### 3.3. Presence of Species Flocks

We found 97 clades with bootstrap support > 60% that included 3 or more Baikal OTUs. Among them, 42 clades (hereafter, potential species flocks) contained Baikal OTUs with the following characteristics: (1) <3% of sequence divergence and (2) the abundance of at least three OTUs within the flock differed < 100 times from each other. Most of these putative flocks were associated with Stramenopiles (Chrysophyceae, Bicosoecids, Bolidophyceae, and Dictyochophyceae), but other supergroups also featured them: Choanoflagellida (Opisthokonta), Pansomonadida (Rhizaria), Cryptophyceae, Prasinophytes (Archaeplastida), Perkinsiides, Ciliata, and Dinoflagellata (Alveolata) (Table 1).

Chrysophyceae was the richest group with 11 such potential flocks in different genera (Figure 5, Appendix A). Most of them included 3–6 Baikal OTUs. However, one flock (92 bootstrap support) contained 53 Baikal OTUs, which displayed 2.2% sequence divergence on average, also including 6 unclassified protists from other locations (one from Arctic Char Lake, two from Columbia River estuary, and one from a shallow French pond). A flock within the group Pansomonadida (Rhizaria) contained 7 Baikal OTUs, with sequences differing from each other by 2.4% on average (Figure 6a). These Rhizarian OTUs may represent an endemic Lake Baikal species flock as their similarity to other known sequences was < 90%. Five potential species flocks within Dictyophyceae, which was previously known in Lake Baikal by one taxon *Pseudopedinella* sp., are shown in Figure 6b.

Lastly, we found 24 clades where the differences between OTUs were > 3%, yet other features (bootstrap support, OTU abundance) were comparable to those observed in the species flocks indicated above. In particular, bicosoecids contained 10 such clades (Appendix A).

### 3.4. Potentially Endemic Baikal OTUs

About 7% of Lake Baikal OTUs had between 80–90% similarity to known sequences. Most of these OTUs were counted > 2 times within samples (from 3 to 1848 reads). According to the phylogenetic analyses, they belong to Stramenopiles (45 OTUs), Opisthokonta (26 OTUs), Alveolates (15 OTUs), Rhizaria (10 OTUs), and Achaeplastida (1 OTU). Most of these potentially endemic OTUs clustered with sufficient statistical support with other taxa. Moreover, some of them were included within the mentioned putative species flocks and possibly originated via radiation. In particular, this is true for the potential flock within Pansomonadida (Rhizaria, Figure 6a) and within Gregarinasina, where four novel alveolate OTUs (totaling 140 reads) formed one clade (Alveolata, Appendix A, ML support values 100). Other potentially endemic Alveolate OTUs belonged to other gregarinasins (2 OTUs), perkinsiids (6 OTUs), dinoflagellates (2 OTUs), and ciliates (1 OTU). Novel Stramenopile OTUs clustered mainly with bicosoecids (37 OTUs, Appendix A), but also with Chrysophyceae (4 OTUs, Appendix A) and MAST (3 OTUs, Appendix A). OTU_153 (totaling 872 reads) had ~85% similarity to the closest reference sequence, which belongs to an unclassified marine organism from Ochrophyta, but it did not cluster with any other sequence in the phylogenetic analysis with sufficient bootstrap support (Appendix A). Potentially novel opisthokonts clustered with Cryptomycota and Chytridiomycota (16 OTUs) and Choanomonada (10 OTUs) (Appendix A). One novel archaeplastid OTU was placed on the long branch within the family Mamiellophyceae (Prasinophytes) (Appendix A).

There were numerous Baikal OTUs that had > 90% similarity to known sequences but did not cluster with them with good bootstrap support (> 60). For example, OTU_154 (totaling 736 reads) was related to the Novel Clade 10 (Rhizaria) but did not cluster with other sequences within this clade (Appendix A). OTU_426 and OTU_327 clustered with Endomixa (Rhizaria) but with low bootstrap support, being located in the long branches (Appendix A). Several relatively common chlorophyte OTUs were related to Trebouxiophyceaea with low support (Appendix A). Overall, such potentially novel Baikal OTUs could be found in most phyla.

### 3.5. Community Composition

OTU richness per sample ranged between 200–500 OTUs with the exception of the community from the Northern basin (3ND(1)), which was more diverse, containing about 700 OTUs (Figure 7a). Community similarity was high (i.e., Bray Curtis dissimilarities below 0.37) in assemblages from similar depths influenced by the Selenga river (1SC and 3SD(1), 1SC and 1CD(1), 2SD(50), and 3SD(50); Figure 7b) as well as in assemblages from the same depths that were sampled from open waters in the South and Central basins (1SD(25) and 4CD(25); Figure 7b). Other assemblages differed substantially from each other (Bray-Curtis average ~0.7). Communities from deeper water layers (25–50 m) were more similar to each other than to those from upper layers, even when they were taken in different geographic locations (Figure 7b).

Many phylotypes were shared between the three Baikal basins (Appendix A). However, members of Bolidophycea, Pelagophycea, and Choanoflagellida were not found in two samples that were taken near the Selenga river estuary (2SD(15) and 3SD(15)). Also, members of Pelagophycea were not found in samples near the Selenga river estuary (1SC and 3SD(1)) and in samples from the North basin (1NC, 2ND(25) and 3ND(1)). Members of Choanoflagellida were absent as well in two Northern samples (1NC and 1ND(15)). The Northern basin contained many unique phylotypes (457 OTUs) due to the highly diverse 3ND(1) sample (Figure 7a, Appendix A).

## 4. Discussion

Most lakes in the world originated during the last glaciation < 20,000 years ago [57]. Therefore, our current knowledge on microbial populations inhabiting these waterbodies derives from research on geologically young habitats. Investigating microbial communities in ancient freshwater lakes such as Lake Baikal, which is ~26 million years old, can provide new insights on the evolution of microbes in freshwaters as well as on their immigration and local extinction history over millions of years [58]. Given that most lakes are younger than the majority of microbial lineages they contain, research on ancient lakes could provide crucial information that would help us understand how freshwater microbiota has evolved during the last million years and what evolutionary dynamics are prevalent.

A recent study using both plankton and benthic Baikal samples coupled with 454 DNA sequencing identified 644 protist OTUs (at 97% sequence similarity) and proposed the existence of novel lineages among parasites [27]. Our results indicate that Baikal protist communities are much richer. Only one fraction of the plankton (0.45–8 μm), collected during one week from a relatively small depth range (1–50 m), contained between ~800 and ~1400 OTUs (at 97% and 99% similarity, respectively). For comparison, there were between 82 to 715 OTUs (at 97% similarity, 0.22–50 μm size-fraction, Illumina sequencing) per lake in one of the latest freshwater-protist metabarcoding studies including 227 Pyrenean mountain lakes [7]. Given its long history and the fact that Lake Baikal could have served as glacial refugia during the last glaciations [59], protists species may have accumulated in the lake more than in other younger waterbodies, thus partially explaining their higher richness. Furthermore, the complex topography of the lake (large bays, vast shoals, deep open waters, geothermal springs) provides multiple ecological niches for resident and immigrant taxa. Rivers also have likely contributed to increased immigration into Lake Baikal (nowadays, ~300 rivers discharge into the lake) by bringing living cells or cysts. In particular, Popovskaya [60] has shown that the northern zone of Lake Baikal has a higher species richness of picophytoplankton due to rivers’ discharge. Concordantly, in our data, the sample from this northern part of the lake (3ND(1)) had the highest richness.

### 4.1. Freshwater and Marine Lineages in Lake Baikal

A substantial fraction of Baikal OTUs clustered with DNA sequences from other freshwater protists. This could be explained by past migrations with Lake Baikal acting as a source and sink of immigrants. Given its potential role as a glacial refugium for freshwater protists along with its huge size, Lake Baikal may have been an important source of immigrants that colonized the newly formed lakes in Northern Eurasia and North America after the last glaciation. These events may partially explain the presence of identical or highly similar protistan OTUs in Lake Baikal and in other younger glacial lakes.

Freshwater origin was established for most studied multicellular organisms in Lake Baikal (e.g., Baikal sponges, Baikal oilfish), but not for all of them. In particular, the neoendemic Baikal seal *Pusa sibirica* diverged between 0.72–1.60 million years ago from a marine Arctic ancestor [61]. Similarly, apart from freshwater protists, we found other protists that are evolutionarily close (≤1% divergence in 18S rDNA) to brackish/marine species from the Baltic Sea or Arctic marine waters. These marine-derived Baikal protists occurred in various lineages, such as ciliates, dinoflagellates, chlorophyta, telonemids, fungi, pelagophycea, and amoebophrya. This is consistent with other results indicating close evolutionary relationships between specific bacteria from Lake Baikal and the Baltic Sea [62] as well as close relationships between under-ice blooming dinoflagellates in Lake Baikal, the Baltic Sea, and the arctic area [27,63].

Our results suggest that a fraction of Baikal protists are phylogenetically close to Arctic protists, and one of the main differences between them would be the salinity of their habitats. How arctic species overcame the salinity barrier [64] may be explained by two reasons. First, the salinity of polar seas fluctuates highly (nowadays and in the past) and could decrease due to riverine discharge and ice melting [65]. Thus, their microbial inhabitants often have a wide salinity tolerance [66] that may allow them to colonize freshwater environments or recolonize marine waters. In particular, we have found wide salinity tolerance in two Arctic ancestors of Baikal dinoflagellates [27,63]. Second, according to Karabanov et al. [59], certain ecological niches in the pelagic zone of Lake Baikal may have been empty during the last cooling period in the Pleistocene due to drastic tectonic and climate change. If this hypothesis is true, this may have promoted new “marine” invaders to settle in the lake. In contrast, Segerstrale [67] suggested that relicts from Siberia, which did not tolerate high salinity, moved through the system of proglacial waters and the White Sea, which was a freshwater lake at that time, to the Baltic Sea. In sum, the planktonic protist assemblages at Lake Baikal seem to contain freshwater as well as species that may have crossed the salinity boundary relatively recently; the latter could represent relicts of the last glaciation. Interestingly, the existence of taxa with close marine relatives in Lake Baikal has already been suggested [68], but such a scenario was not popular among scientists. Our results suggest that the previous hypothesis could have more relevance for protists than for multicellular organisms.

Additionally, we found certain high rank taxa in Lake Baikal that are considered predominantly marine (Figure 4). In particular, we found freshwater representatives of the group Pelagophyceae, which were phylogenetically close to other Pelagophyceae from the Beaufort Sea (Arctic). Marine groups such as Labyrinthulomycetes, various MAST, pirsoniids, and Bolidophyceae were found in other studies both in ephemeral freshwater bodies [12,69], in large shallow eutrophic [16] and oligotrophic [10] lakes, and, as reported here, in the ancient and stable oligotrophic waters of Lake Baikal. This suggests dispersal but no long-term colonization (i.e., in ephemeral waters) as well as cases of long-term colonization via adaptive evolution.

### 4.2. Did Lake Baikal Protists Radiate?

Adaptive (and possibly non-adaptive) radiation is one of the main mechanisms responsible for the ecological and phenotypic diversity of multicellular life [70,71]. Yet, much less is known about the operation of this process in protists. Recent radiation has been proposed for dinoflagellates in Lake Baikal [26] as well as for a diatom species flock in Lake Ohrid [72]. Furthermore, the existence of species flocks in diatoms has been extensively discussed in a number of ecosystems [73,74,75]. In addition, based on metabarcoding data, novel radiations within Lake Baikal fungal parasites were suggested [28].

According to the definition of Schluter [47], adaptive radiations are characterized by four features in the resulting species flock: (1) common ancestry, (2) rapid speciation, (3) phenotype-environment correlation, and 4) trait utility. Usually, radiated species are endemic to a specific region, but this is not a mandatory feature [71]. The previous should be especially true for protists, which have high dispersal abilities. Here, we investigated 42 bootstrap-supported clades, which contained close (>97% identity on average) but not identical Baikal OTUs (at 99% similarity). We suggest that at least some of them may be regarded as species flocks that originated via radiation, though it is impossible to check the two last adaptive requisites mentioned previously using DNA data [47]. The reasons supporting these species flocks are: (1) these OTUs constituted monophyletic clades with >60% bootstrap support and their similarity indicated that they are closely related, yet distinct, taxa; (2) OTUs were located on relatively short branches, suggesting that all species within the flock may have originated over a short period of time. Most of our potential species flocks contained 3 to 10 OTUs, however one Chrysophyceae flock contained 53 OTUs. Moreover, Chrysophyceae was the taxon featuring most potential flocks (Table 1, Figure 5). This agrees with results from [76], who proposed, based on SEM microscopy, that Lake Baikal is a hotspot of silica-scaled Chrysophyte diversity.

An alternative explanation for the previous could be intragenomic or intercellular variation [48]. However, concerted evolution should eliminate different copies of the rRNA gene within genomes in evolutionary times [77]. Still, certain protist lineages may keep different copies of the 18S rDNA in their genomes [78,79]. Indeed, we found 192 OTUs in Lake Baikal, which were similar to the dinoflagellate *Gyrodinium helveticum*. Among them, one OTU was abundant (214,236 reads in total) and identical to *G. helveticum* from Lake Baikal [80], while others were represented by < 100 reads. Potential intragenomic variability was also evidenced in Katablepharidaceae: seven OTUs were included in a well-supported clade, but one of them was found 30,958 times, being present in all samples. OTUs that were similar to it were much rarer (2–86 reads). A similar example was found within Cryptophyceae. Apart from the previous cases suggesting intragenomic variation, we are still left with multiple examples that can be regarded as species flocks (Table 1).

In addition to the cases described above, we found monophyletic clades where the difference between Baikal OTUs was > 3%. It is hard to ascertain if their origin evolution was progressive or if they represent ancient radiations. Also, as our analytical approach was based on the conserved 18S rRNA gene, we were not able to consider the most recent radiations, which may have happened < 1 million year ago, when a fraction of Baikal neoendemics, including protists, originated [27,63,81,82]. Nevertheless, we found potential species flocks in diverse lineages (Table 1), suggesting that evolutionary radiations could be as important for single-cell eukaryotes as for multicellular counterparts [20,82].

### 4.3. Potential Endemism within Baikal Protists

The amount of endemism in microbiotas is a matter of debate [83]. Large ancient lakes can provide novel insights on the previous discussion given that different to most lakes, microbes have had the time to diversify in them [75]. Microbial endemism in Lake Baikal has already been suggested based on scanning electron microscopy for certain relatively large protists [24,27,73]. We studied the community of small protists, featuring 0.4–8 µm of cell size, which should have a high dispersal potential [3]. Nevertheless, we found several types of 18S rDNA sequences that could belong to endemic organisms. These are: Type 1—OTUs that seem to have originated in the lake via radiation, Type 2—OTUs displaying < 90% similarity to other known protists, and Type 3—OTUs that did not fall into clades with sufficient statistical support. These potentially endemic OTUs were obtained from different samples and passed our strict quality filters, thus they should not constitute errors, but represent true diversity.

A key question is whether macro and micro endemics in Lake Baikal emerged via similar evolutionary mechanisms. Despite the large size of Lake Baikal, there is limited evidence of geographic isolation promoting within-lake diversification. Sympatric evolution was invoked as the mechanism explaining the diversification of some multicellular organisms in the lake [84]. In particular, this explanation was used for Baikal actively-swimming coregonid fishes [85] and cottoid fishes [86]. Overall, studies of macroorganisms invoke two main scenarios to explain the origin of Baikal endemics: rapid radiation from one ancestor (e.g., sponges [87]) and endemic species emerging after multiple invasions (repeated colonizations) from one lineage (e.g., isopods [88]). According to our results, a fraction of the protistan lineages populating Lake Baikal appear to have experienced diversification processes that are coherent with these scenarios: the first scenario would be represented by the Type 1 OTUs and the second by a fraction of Type 2 and Type 3 OTUs.

A good example of endemism in Lake Baikal is likely represented by parasitic Gregarinozina (Apicomplexa). Six Baikal Gregarinozina species are known based on microscopy [24]. All of them are endemic parasites of endemic amphipodes and three parasite species seem to belong to a species flock. We found seven OTUs belonging to Gregarinozina. Among them, four OTUs differed from other known DNA sequences by 11.8–13.7% and clustered into a putative species flock. Also, two additional OTUs differed from other known sequences by 6.2% and 13.0%. Thus, a total of six potentially endemic Baikal Gregorinozina including one species flock were independently predicted by microscopy and DNA metabarcoding. We found that about 25% of unique Baikal OTUs belonged to parasites, which may also represent endemic taxa adapted to endemic hosts. A prominent part of other potential endemic protists belonged to organisms (e.g., bicosoecids and choanomonads), which are commonly attached to substrates (particles or another organism), a lifestyle that could promote endemicity. For example, Jankowski [89] have shown that Baikal epibiont ciliates, which attach to various macroorganisms, show high levels of endemism in contrast to free-living ciliates. Similarly, tiny attached flagellates may follow comparable trends: we found diverse bicosoecids in site 3ND1, which includes abundant aquatic plants that could be their hosts.

Overall, the potential endemicity of small Baikal protists may be associated more with ecological or physiological features developed by local adaptation than with dispersal limitation. Indeed, there is no evidence of a strong geographic barrier for organisms from Lake Baikal as a mix of palearctic and Baikal taxa are present in various parts of the lake [18]. Moreover, a few endemic animals and microalgae from Baikal dispersed to some Siberian rivers and lakes [18,90]. A prominent example is illustrated by Baikal gammarides: the genus *Pallasea* consists of Baikal endemics, except *Pallasea quadrispinosa*, which is a close relative of Baikal species but expanded to North and Central Europe during the last glaciation [90]. In contrast, many endemic gammarids, with specific ecological requirements, did not disperse out of Baikal [18].

### 4.4. Community Structure of Small Baikal Protists

Baikal piconanoplankton has been predominantly associated with cyanobacteria and “some tiny flagellates”. Our approach allowed us to identify some of these “tiny flagellates” as highly diverse bicosoecids, choanoflagellates, specific cercozoa (like Novel clade 10), bolidophyceae, pelagophyceae, telonemids, perkinsiids, and fungi-like parasites. Previously, it was shown that *Nannochloropsis limnetica* from Eustigmatophyceae plays a prominent role in Baikal picoplankton [91]. Indeed, this family was represented by many reads in most samples and corresponded to two different OTUs (one was *N. limnetica* and another one was close to *Monodopsis subterranea*).

It has been shown that the structure of Baikal microbial communities in spring is affected by lake topography [28]. In our study, summer communities differed from each other, but we did not observe a clear correlation with the lake’s topography. However, we found that Baikal communities may be influenced by riverine inputs as surface communities near the delta of the Selenga river clustered together. Also, the community from site 3ND (Figure 7a), which is under the influence of two big rivers, had a higher species richness than the rest of the samples, pointing to community coalescence [92]. In addition, we found that depth influenced community composition as surface communities differed from those at 25–50 m depth. Previously, it was shown that picophytoplankton decreased in the 20–30 m samples compared to the 5 m depth samples in July [91].

## 5. Conclusions

Lake Baikal is one of the most important lakes in the world considering its long ecological history and economic relevance. Its study can help to understand the evolution and community structure of lacustrine microbiotas around the world, as well as their susceptibility to global change. Our results indicate that the richness of Lake Baikal protists is higher than in other lakes in temperate zones, which may reflect Lake Baikal’s long history or its role as a glacial refugium. In addition, we found that Lake Baikal protist microbiota contains a number of marine-derived species, pointing to relatively recent colonizations of its fresh waters by some brackish or marine protists. Moreover, we found potential examples of evolutionary radiations within the lake that agree with other radiations reported for multicellular organisms in Lake Baikal. OTUs corresponding to potential radiations and OTUs showing a high differentiation (>90%) from known sequences constituted about 23% of all OTUs and could represent endemic Baikal protists. If future studies confirm these putative radiations, then Lake Baikal will become a paramount example of evolutionary radiations across the macrobial and microbial worlds.

## Figures and Tables

**Figure 1 microorganisms-08-00543-f001:**
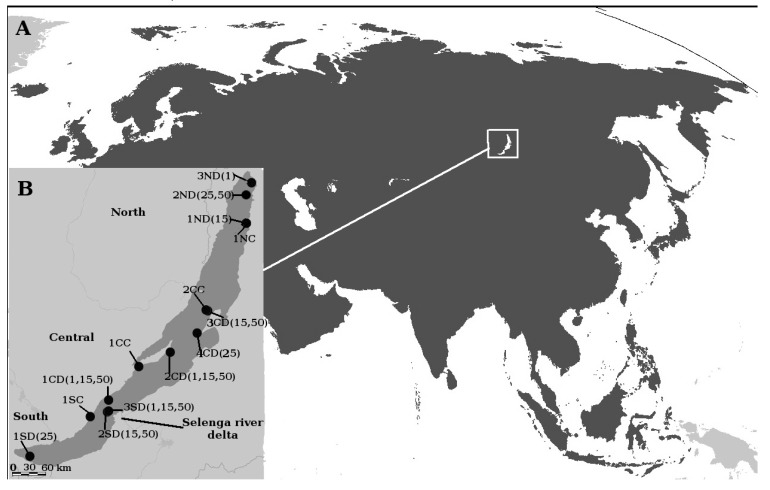
A. Lake Baikal position in Eurasia, B. Sampling sites in Lake Baikal. For each sample, the first number and letter indicates the number within one of the basins (S—Southern, C—Central, N—Northern basins), the second letter indicates coastal (C; up to 40 m depth) or deepwater (D; more than 100 m depth) sites, and numbers within brackets indicate the sample depth (e.g., 1CD (1, 15, 50) means of samples from 1, 15, and 50 m depths that were taken in the first site of the Central basin at a deep site. No depth is shown for the coastal samples.

**Figure 2 microorganisms-08-00543-f002:**
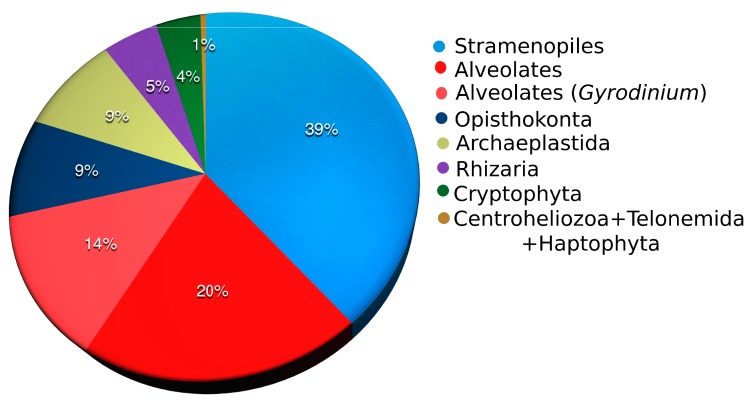
Relative abundance of protist high-rank groups in Lake Baikal.

**Figure 3 microorganisms-08-00543-f003:**
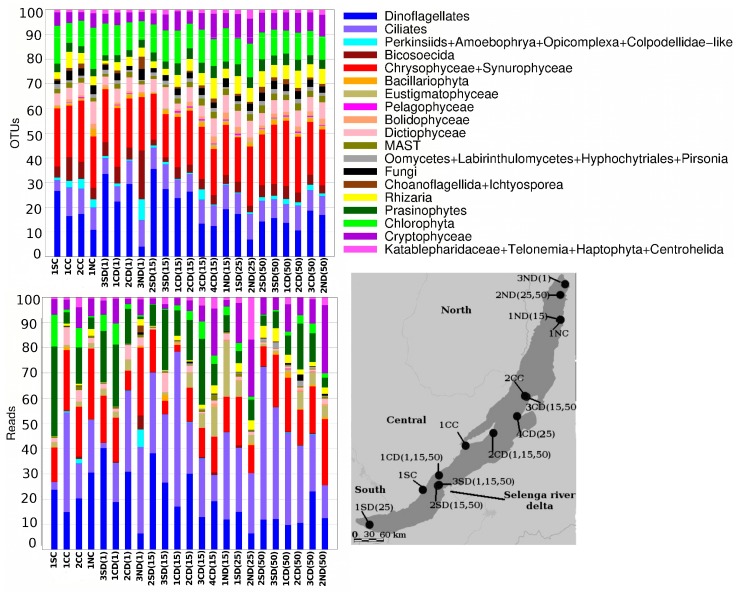
Histograms showing the relative proportion of different OTUs (**a**) and 18S rDNA amplicon reads (**b**), assigned to high-rank taxa across different sample sites.

**Figure 4 microorganisms-08-00543-f004:**
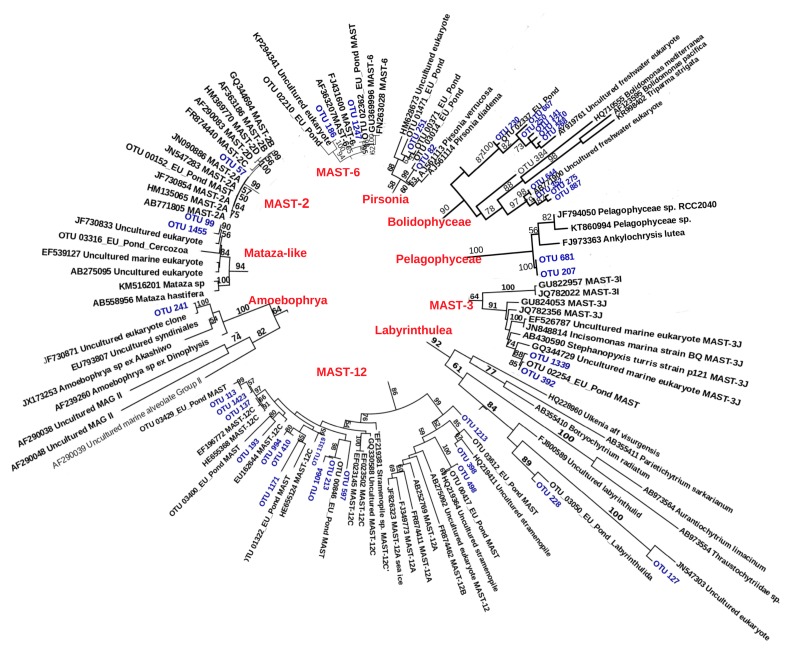
Sections of Maximum Likelihood (ML) trees representing clades that are known mostly from marine waters and that include Baikal OTUs (in blue) (full versions of the trees are included in Appendix A). Numbers next to branches indicate bootstrap support.

**Figure 5 microorganisms-08-00543-f005:**
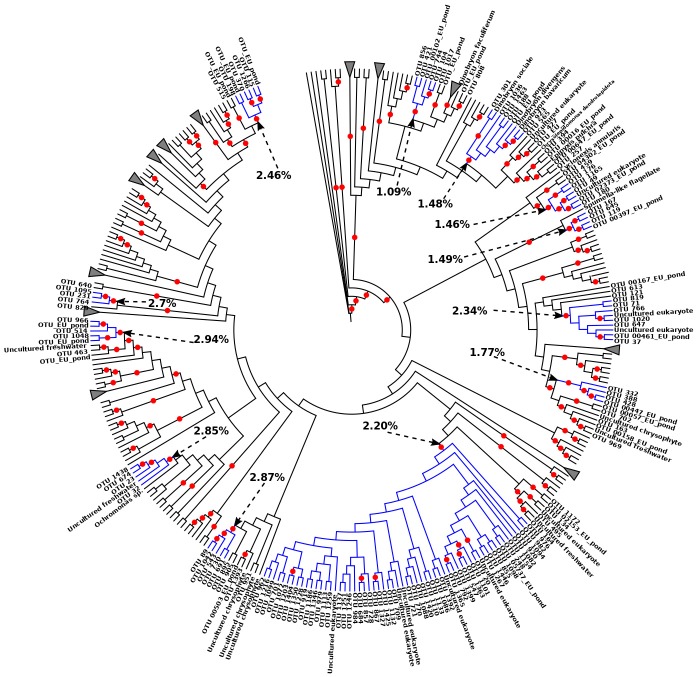
Reduced Chrisophyceae ML tree (See full version of the tree in Appendix A). Potential species flocks are indicated in blue. The percentages indicate the average divergence of the sequences within each flock. Red circles correspond to bootstrap supports > 60.

**Figure 6 microorganisms-08-00543-f006:**
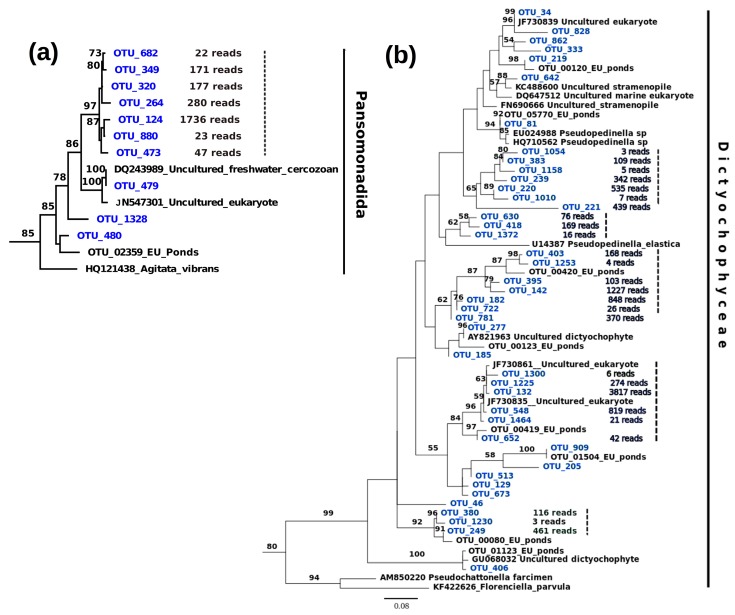
(**a**) Potential species flock in Pansomonadida (Rhizaria), which includes 7 OTUs differing > 10% from any known DNA sequence; (**b**) Potential species flocks in Dictyochophyceae (see full tree in Appendix A). Baikal OTUs are shown in blue. The number of total reads, considering all samples, are shown for each Baikal OTU from the flocks. Flocks are shown with a dashed line.

**Figure 7 microorganisms-08-00543-f007:**
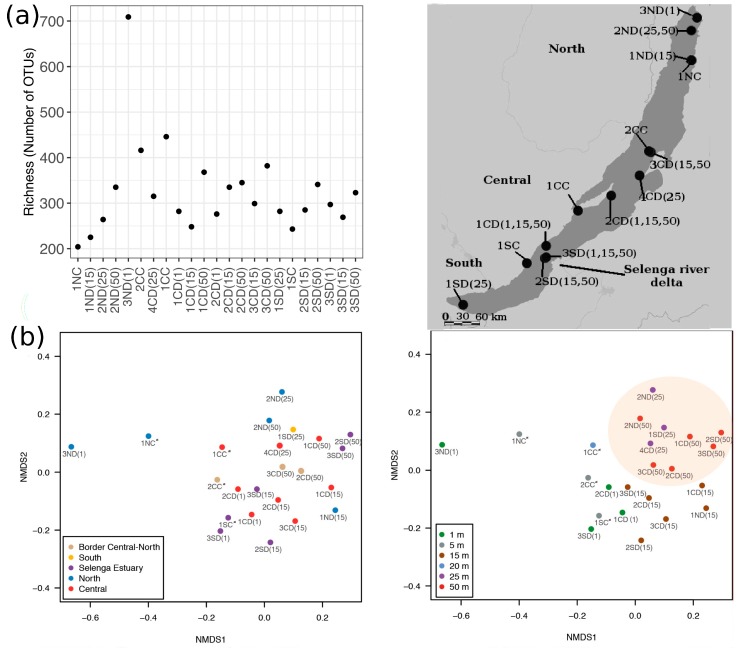
(**a**) OTUs richness in different samples (left) and the map, showing the location of the samples (right). (**b**) NMDS analyses indicating the variable composition of protist assemblages within different basins (left) and different depths (right). The orange oval on the right indicates samples from deeper waters (25 and 50 m). Coastal samples are marked with an asterisk.

**Table 1 microorganisms-08-00543-t001:** Potential species flocks in Lake Baikal.

Phylum	Taxon	Number of Potential Species Flocks
Stramenopiles	Chrysophyceae	11
	Dictyochophyceae	5
	Bicosoecida	4
	Bolidophyceae	2
Alveolata	Ciliata	4
	Perkinsiidae	5
Opisthokonta	Choanoflagellatea	2
	Chytridiomycota	2
	Rozellida	1
Archaeplastida	Chlorophyta	4
Rhizaria	Thecofilosea	1
	Sarcomonadea	1
	Novel Clade 10	1
Hacrobia	Cryptophyceae	1

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
