# Peer review of "Tracing the Origin of Planktonic Protists in an Ancient Lake"

_microorganisms, 2020, doi:10.3390/microorganisms8040543_

Round 1

Reviewer 1 Report

The manuscript was significantly improved. 

L. 148-149. "Those OTUs that differed between each other 1% or less were regarded as similar. Then, OTUs that displayed >10% difference from any known DNA sequence were regarded as potential endemics of the Baikal region". How do you define OTUs with a similarity level between 1 and 10% difference? This should be specified. 

L. 152. The criteria for identification of species flocks should be specified in the Methods section. 

After clarifying of these 2 questions the Manuscript can be accepted. 

Author Response

Comment: The manuscript was significantly improved. 

Answer: we thank the reviewer for his/her time to review our work and the comments suggestions to improve it.

Comment:L. 148-149. "Those OTUs that differed between each other 1% or less were regarded as similar. Then, OTUs that displayed >10% difference from any known DNA sequence were regarded as potential endemics of the Baikal region". How do you define OTUs with a similarity level between 1 and 10% difference? This should be specified."

Action: We have amended this section including the required information. The section now reads:

"OTUs that differed from known DNA sequences ≤ 10% were considered as taxa with unclear distribution: they could belong to both widespread or endemic organisms. Those OTUs that differed ≤1% from other reported sequences were regarded as similar. Then, OTUs that displayed >10% difference from any known DNA sequence were regarded as potential endemics of the Baikal region."

Comment: L. 152. The criteria for identification of species flocks should be specified in the Methods section. 

Action: We have included the required information in this section, which now reads:

"To search for putative cases of species radiations we used the following criteria. Three or more 99%-similarity OTUs should form one clade with bootstrap support > 60%. OTUs from such clades should be similar (>97% identity in average). These criteria aimed to reflect features of adaptive radiations such as common ancestry and rapid speciation [75]. Also, the abundance between the two most abundant OTUs within such clades should differ <100 times, otherwise the variation was regarded as intragenomic."

After clarifying of these 2 questions the Manuscript can be accepted. 

Reviewer 2 Report

I have no further comments.

Author Response

We thank the reviewer for his/her time to evaluate our work and the suggestions to improve it.

This manuscript is a resubmission of an earlier submission. The following is a list of the peer review reports and author responses from that submission.

Round 1

Reviewer 1 Report

The manuscript of Annenkova  et al. describes the planktonic protists communities in the surface water of an ancient lake in Russia, Lake Baikal. I have the following comments:

1. The authors should include the country of Lake Baikal, Russia, in the abstract.

2. Some sentences in the manuscript are bold, with different font size/type, please check and modify.

3. Line 154, what are the reads distribution across the 23 water samples?

4. Line 155, why did the authors use a cut-off value of 99% sequence identify?

5. Line 180, although this is not the focus of the manuscript, the authors should also mention the subdivisions/classes of fungi in the samples, to be in consistent with those in Line 179.

6. Please include the full name of MAST, Marine stramenopile , in the manuscipt.

Reviewer 2 Report

The article is well written and presents interesting data on an important group of organisms and the important problem of endemism in microorganisms. The article contains important information, but authors need to present their results in a broader research context. The article has three major drawbacks.

Firstly, it is not clear which size class of microeukaryotes has been studied. The authors used a preliminary filter with a size of 70 microns and 2 filter sets with a pore size of 8 microns and 0.45 microns (lines 97-98), but it is not clear from the text whether these two types of filters were analyzed together or separately. If these filters were analyzed separately, then you need to provide data and discuss the similarities or differences between the two size classes.

Secondly, this is the lack of a comparative analysis of the data obtained in the general context of the endemism of the Baikal biota. The existence of biogeographic populations is largely related to the size of organisms, which is associated with the organism's potential for dispersion. The study was conducted on a specific size class and it is necessary to compare the data obtained with the results obtained for other size classes. Partially, these data are given in reference to Timoshkin's article on lines 52-53, but the authors need to supplement the discussion and focus their attention on this aspect.

Thirdly, in the abstract or conclusion there is no clear summary of the study. What percentage of OTUs have a cosmopolitan distribution according to the chosen level of similarity? What percentage of OTUs may be potentially endemic? What types of organisms are predominantly endemic and which are mainly cosmopolitan? Is this related to the peculiarities of their size or physiological characteristics? Partially the answers to these questions are scattered in the results and discussion section, they need to be collected and presented in a concise form in the abstract and conclusion.

I recommend a Major revision.

Minor comments:

1) There is a contradiction between lines 51-53 and 61-62.

L.51 Baikal present-day communities inhabit the whole water-column and include more than 3,500 known species of eukaryotic organisms, of which more than half are endemic [18].

L.61 Compared to its eukaryotic macrobiota, much less is known on the diversity, evolution, and ecology of Lake Baikal eukaryotic microbiota.

The first sentence should be clarified whether it is related to all eukaryotes in lake Baikal or only eukaryotic macrobiota. From the second sentence, we can understand that eukaryotic microbiota is poorly studied, but the first sentence means that we already have enough information on microbiota to make conclusions about their endemicity. 

2) L. 75-81. Format text

3) L. 82. “was” instead of “is”. The next sentence is in the Past tense and the aim of the study should precede the results.

4) L. 89. “Niskin” instead of “niskin”. After the name of Shale Niskin.

5) L. 90. Space between “40m”

6) L. 123. “Python” instead of “python”

7) L. 123. “BLAST hits were filtered prior to taxonomy assignment using an in-house python script”. No reference to the script. The script should be published.

8) L. 129-130. Web-reference should be formatted, now it doesn’t work properly. “accession numberS”? There is only one number. “Sequences are publicly available at the European Nucleotide Archive (http://www.ebi.ac.uk/ena; accession numbers PRJEB24415).”  URL is not allowed in the text of the MDPI article, provide it in the reference list

9) Sample collection section. Missing information about the number of replicates of samples, filters from one sample, number of amplifications from one extract etc.

10) L. 131-134. The statistics is described poorly. For example, what was the cut-off value for high or low similarity et c.? It should be described here.

11) L. 169-173. Format text

12) L. 229-231. It is not of polar origin if it is already present in Tianshan. You can describe it as originating from cold environments or alpine and polar environments.

13) L. 351-354. "most of them could be traced back to Arctic waters or the Baltic Sea. "

The presence of a close relative from a certain region does not automatically mean the origin of the phylotype from this region. Now the wording of the proposal directly indicates the origin of the phylotype from the Baltic Sea region, but the available data are insufficient for such a statement. The phrase should be rewritten in such a way as not to mislead readers.

14) L. 359-361. "Second, certain ecological niches in the pelagic zone of Lake Baikal were empty during the last cooling period in the Pleistocene due to drastic tectonic and climate changes [55]"

An interesting theory proposed by Karabanov and co-authors, but the phrase should be rewritten in the article. Now the article directly states that the ecological niche was empty, whereas now we can only assume that it was empty.

15) List of references – double numbering of references